# Cu Pillar Electroplating Using a Synthetic Polyquaterntum Leveler and Its Coupling Effect on SAC305/Cu Solder Joint Voiding

**DOI:** 10.3390/ma17225405

**Published:** 2024-11-05

**Authors:** Wenjie Li, Zhe Li, Fang-Yuan Zeng, Qi Zhang, Liwei Guo, Dan Li, Yong-Hui Ma, Zhi-Quan Liu

**Affiliations:** 1Shenzhen Institute of Advanced Electronic Materials, Shenzhen Institute of Advanced Technology, Chinese Academy of Sciences, Shenzhen 518055, China; 2School of Material Science and Chemical Engineering, Harbin University of Science and Technology, Harbin 150040, China; 3Yantai Research Institute of Harbin Engineering University, Yantai 264006, China; 4Guangdong-Hong Kong-Macau Joint Laboratory for Photonic-Thermal-Electrical Energy Materials and Devices, Institute of Applied Physics and Materials Engineering, University of Macau, Avenida da Universidade, Macao SAR 999078, China; 5Shenzhen College of Advanced Technology, University of Chinese Academy of Sciences, Shenzhen 518055, China

**Keywords:** packaging substrate electroplating, Cu pillar bumps, polyquaterntum leveler, additive adsorption, interfacial reaction, Kirkendall voiding

## Abstract

With the advancement of high-integration and high-density interconnection in chip manufacturing and packaging, Cu bumping technology in wafer- and panel- level packaging is developed to micrometer-sized structures and pitches to accommodate increased I/O numbers on high-end integrated circuits. Driven by this industrial demand, significant efforts have been dedicated to Cu electroplating techniques for improved pillar shape control and solder joint reliability, which substantially depend on additive formulations and electroplating parameters that regulate the growth morphology, crystal structure, and impurity incorporation in the process of electrodeposition. It is necessary to investigate the effect of an additive on Cu pillar electrodeposition, and to explore the Kirkendall voids formed during the reflowing process, which may result from the additive-induced impurity in the electrodeposited Cu pillars. In this work, a self-synthesized polyquaterntum (PQ) was made out with dual suppressor and leveler effects, and was combined with prototypical accelerator bis- (sodium sulfopropyl)-disulfide (SPS) for patterned Cu pillar electroplating. Then, Sn96.5/Ag3.0/Cu0.5 (SAC305) solder paste were screen printed on electroplated Cu pillars and undergo reflow soldering. Kirkendall voids formed at the joint interfaces were observed and quantified by SEM. Finally, XRD, and EBSD were employed to characterize the microstructure under varying conditions. The results indicate that PQ exhibits significant suppressive and levelled properties with the new structure of both leveler and suppressor. However, its effectiveness is dependent on liquid convection. PQ and SPS work synergistically, influencing the polarization effect in various convective environments. Consequently, uneven adsorption occurs on the surface of the Cu pillars, which results in more Kirkendall voids at the corners than at the center along the Cu pillar surface.

## 1. Introduction

With the advancement of high-integration and high-density interconnection in chip manufacturing and packaging, Cu bumping technology in wafer- and panel- level packaging is being developed for micro-sized structures and pitches to accommodate increased I/O numbers on high-end integrated circuits [1]. Cu pillar bumps are primarily fabricated using photolithography and electrodeposition. In the Cu pillar process, factors such as additive concentration, the convection state of the plating solution, current density, and electroplating equipment influence the flatness of the Cu pillar [2]. Numerous studies have been conducted on this process, including electroplating Cu pillars at different current densities to achieve desirable properties [3] and utilizing a new type of plating bath to produce flat Cu pillars [4]. A Cu electroplating bath typically involves Cu(II) sulfate hydrate, sulfuric acid, and hydrochloride acid, and, most importantly, organic additives classified as suppressors, accelerators, and levelers based on their electrochemical properties. Driven by the above-mentioned industrial demand, significant efforts have been dedicated to Cu electroplating technique for improved pillar morphology control and solder joint reliability [5]. Li et al. [6] studied a series of polyethylene glycol (PEG)-based molecules of tailored terminal groups as suppressors for the profile control of Cu pillars. Their inhibitory capabilities varied due to steric hindrance differences originating from functional groups and alkyl lengths. Additionally, when the molecular weight is less than 1000, the effect of the additive PEG with terminal functional group on Cu inhibition is greater than that of simply increasing the molecular weight. Pasquale et al. [7] analyzed the effects of chloride in Cu pillar electroplating. Surprisingly, it was found that the absence of chloride results in prototypical accelerator 3-mercaptoyl-2-propyl sulfonic acid (MPSA) and weak inhibition of the suppressor polyethylene glycol (PEG) on Cu surface, demonstrating the pivotal role of chloride in synergetic interactions among additives [8]. 

Understanding additive adsorption and interactions under multi-physical fields are critical to high-performance Cu electroplating, and numerous researches have been conducted [9,10]. Zhu et al. [11] used Janus green B (JBG) and a commercial leveler to electrodeposit Cu pillars. Mass spectrometry analyses revealed that this leveler incorporated a significant number of contaminants into the Cu electroplating, therefore leading to increased Kirkendall voids at the solder joint interface after high-temperature storage. The Kirkendall void formation is a critical reliability concern for solders joints in electronic packing. 

Micro-bumps in three-dimensional integrated circuits (3D ICs) have emerged as a crucial technology. As the miniaturization of power devices approaches its limits, 3D packaging technology offers a viable solution for achieving higher density. However, this advancement has also led to significantly heightened requirements for micro-bumps [12]. The electrodeposition process frequently yields a highly reliable solder layer. Szostak et al. [13] reported that through the electrodeposition of Au-Sn eutectic, a distinct pattern is formed, resulting in a bonding strength exceeding 70%. However, usually, improving the solder joint reliability of electroplated Cu materials substantially depends on additive formulae and electroplating parameters, which determine the growth morphology, crystal structure, and impurity incorporation in the process of electrodeposition and electroplating parameters [14,15]. Chiang et al. [16] reported that Cu electroplated films are impurities affected. After thermal 200 °C aging, many voids appear in the interface due to the huge sum of impurities incorporated in the Cu electroplated films. The control of additives and the quantity of impurities in Cu electroplated films significantly affect void formation at the joint interface. However, investigations on these factors are relatively rare. 

In this work, a self-synthesized polyquaterntum (PQ) was screened out of dual suppressing and leveling effects, and studied in combination with prototypical accelerator bis- (sodium sulfopropyl)-disulfide (SPS) for patterned Cu pillar electroplating. After the electrodeposition process, SAC305 solder was printed onto the surface of the electrodeposition Cu pillars by screen printing. Subsequently, reflow soldering was conducted to create solder joints. The fully formed solder joint underwent an aging experiment inside a furnace. Upon completion of the aging experiment, the cross-sectional morphology was scrutinized and analyzed. The results indicate that PQ is a promising leveler that displays robust adsorbing activity and suppressing stability on cathode, which enables a flat and uniform deposit profile at a high current density of 5 ASD. Moreover, the Kirkendall voiding of SAC305/Cu solder joints exhibit some interesting distributions along the cross-section, which can be related to the different adsorption behaviors and strengths of PQ there.

## 2. Experimental Procedure

### 2.1. Electrochemical Measurements

Inorganic basic solutions (VMS) prepared for all Cu electroplating comprised 120 g/L CuSO_4_∙5H_2_O, 100 mL/L 98 wt.% H_2_SO_4_, and 50 mg/L 37 wt.% HCl. SPS (Shanghai Aladdin, Shanghai, China) was added as the accelerator and a self-synthesized polyquaterntum (PQ) was selected as the leveler, whereas suppressors like PEG or its analogues were excluded. All chemicals employed were not subjected to further purification.

Electrochemical analyses, including cyclic voltammetry (CV), linear sweep voltammetry (LSV), and chronopotentiometry (E-t) measurements, were performed at room temperature on a potentiostat (Metrohm PGSTAT302N) and with a three-electrode cell. A platinum rotating disk electrode (diameter: 3 mm) was applied as the working electrode (WE, controlled at a rotating speed of 1000 rpm for all tests), a platinum foil (size: 10 × 10 mm^2^) was designated as the counter electrode (CE), and a saturated Hg/Hg_2_SO_4_ (MSE) was used as the reference electrode (RE). In CV tests, the potential was scanned from −0.60 to 0.15 V at a scan rate of 20 m Vs^−1^. In the LSV tests, the potential was scanned from −0.30 to −0.80 V at a scan rate of 1 m Vs^−1^. In E-t tests, a constant current density of 5 ASD was applied to the WE and a certain dose of accelerator (0 to 40 mL/L) or leveler (0 to 200 mL/L) was injected every 240 s. In addition, to assess the issue of convective adsorption, experimental controls were conducted using LSV and E-t speeds of 100 rpm and 1000 rpm. ∆*η* is defined in Formula (1) by subtracting the potential difference measured at 1000 rpm from the potential measured at 100 rpm [17]:(1)$$\Delta \eta =V_{100\;\mathrm{rpm}}-V_{1000\;\mathrm{rpm}}$$

### 2.2. Electroplating and Thermal Aging

Sample electroplating was carried out in a 500 mL polytetrafluoroethylene cell attached to a direct-current power supply (Keithley 2200-20-5) with the above-mentioned VMS and additives. Reciprocal magnetic stirring was applied between the anode of phosphor–Cu plate anode (0.035~0.065 wt.% P) and the cathode of patterned Cu packaging substrate (dry film thickness: 30 μm). Electroplating was conducted at a current density of 5 ASD and lasted for 30 min to obtain a Cu thickness of 30 μm.

Sn96.5/Ag3.0/Cu0.5 (SAC305) solder paste was applied to the electroplated Cu column by screen printing. The solder joint was then produced by reflow for 60 s in a 250 °C furnace. Subsequently, the entire solder joint was subjected to thermal aging tests in the thermal aging furnace. It was aged at 150 °C for 96 h and 168 h, and at 200 °C for 96 h and 168 h.

### 2.3. Material Characterization

The surface/cross-sectional morphologies of electroplated Cu pillars and IMC/voids observation of thermally aged SAC305/Cu solder joints were examined by scanning electron microscopy (SEM, Thermo Scientific Apreo 2, Thermo Scientific, Waltham, MA, USA), using secondary electron and back-scattering electron imaging modes, respectively. The crystal structures of electroplated Cu were measured by X-ray diffractometer (XRD, Bruker D8 Advance), equipped with Cu Kα radiation (λ = 1.5418 Å). The texture coefficients (*TC*) were determined through Equation (2). where *I*, the diffraction intensity of X-ray diffraction, was obtained by experiment; *I*_0_ is the diffraction intensity on the standard card; and *n* is the number of peaks on the diffraction pattern:(2)$${TC}_{\left(hkl\right)}=\frac{I_{\left(hkl\right)}/I_{0(hkl)}}{1/n\sum I_{\left(hkl\right)}/I_{0(hkl)}}$$

## 3. Results and Discussion

### 3.1. Electrochemistry

We developed a quaternary ammonium salt additive. It has a new additive structure as well as that of the leveler and suppressor. As depicted in Figure 1a, the PQ structure of the additive comprises two parts: the benzene ring structure of the positively charged quaternary ammonium cation, and the ether. PEG is a very typical class of inhibitors whose main structure is an ether [18]; quaternary ammonium salt cationic additives also have excellent inhibition and leveling effects [19]. Jo et al. [20] again suggested that the hydrophobic tail of the quaternary ammonium salt inhibits Cu electrodeposition, while the cationic portion adsorbs the anions on the surface of the Cu and thus reduces the roughness of the Cu electrodeposition. To achieve an additive with suppressing inhibition and leveling effect, we therefore developed a quaternary ammonium salt additive. This additive is a combination of the ether of PEG, and the benzene ring and cation in the quaternary ammonium salt. To explore the role of PQ in the solution, the polarization or inhibition ability of PQ was studied using LSV and CV. In the CV curves shown in Figure 1b, strong polarization is observed during cathodic scans. Regardless of the presence of PQ in the solution, the initial electrodeposition potential is around −0.42 V. In the absence of PQ, the initial electrodeposition potential is −0.142 ASD. The addition of PQ reduces the electrodeposition current, concentrating around −0.055 ASD. This reduction is confirmed by a significantly decreased Cu dissolution peak in the CV scan, which starts at −0.42 V and ends at −0.22 V. Without PQ, the dissolution peak value is 8.23 ASD. With the addition of PQ, the dissolution peak becomes narrower and decreases, reflecting polarization in a smaller area. Specifically, the dissolution peak is 0.925 ASD for a solution containing 10 mL/L PQ and 0.57 ASD for a solution containing 200 mL/L PQ, as shown in Figure 1b. Albeit some minimal differences, these CV curves are almost overlapped regardless of PQ concentration, because of the stable and saturated adsorption of it at small overpotentials. Therefore, this indicates that PQ effectively inhibits the crystal growth of Cu.

We conducted tests using LSV and E-t under varying conditions: strong forced convection at 1000 rpm and weak forced convection at 100 rpm. We simulated the convection conditions at the outer ring and top center of the Cu pillar at these respective rotation speeds. In the LSV curves (Figure 1c,d), as 10 mL/L PQ is introduced, the onset potential for Cu electrodeposition is negatively shifted and the cathodic current for Cu reduction is prominently reduced. The electrodeposition of Cu is inhibited by PQ. The cathode is monotonically polarized with the continuous addition of PQ up to 200 mL/L, showing the strong polarizing effect of PQ and controllable Cu electrodeposition over a wide overpotential range. However, under conditions of strong and weak forced convection, distinct differences are observed in the LSV curves. In the LSV curves obtained at 1000 rpm and 100 rpm (Figure 1c,d), the following observations are made: at 100 rpm, the initial onset potential (Eonset) without the additive is −0.51 V. With the addition of PQ, polarization intensifies; at 200 mL/L PQ, Eonset shifts to −0.60 V, resulting in a ΔEonset of 0.09 V. At 1000 rpm, the Eonset without PQ is −0.50 V. As the amount of PQ increases, the degree of polarization also increases. With 200 mL/L PQ, the electrodeposition potential becomes −0.65 V at 1000 rpm, and the ΔEonset is 0.15 V. These results indicate that weak forced convection mitigates the monotonic polarization effect of PQ in the solution, narrowing the overpotential range compared to strong forced convection. Based on the LSV results, convection-dependent adsorption (CDA) was observed in PQ: stronger forced convection of the electrolyte led to increased adsorption and inhibition of the leveling agent. The inhibition of PQ was particularly pronounced in environments with strong forced convection. Lee et al. [21] reported that under conditions of strong and weak forced convection, there is variation in the adsorption of quaternary ammonium salt additives, which influences their suppressive effects. The difference in polarization resulting from changes in flow rate can be attributed to the rapid reaction of the quaternary ammonium cation with Cu, which produces an inhibition effect in high-flow conditions. Conversely, at a slow flow rate, such as 100 rpm, the diffusion of additives is sluggish, thereby limiting their effectiveness. As a result, the polarization interval at 100 rpm is narrower than that observed at 1000 rpm. 

In the curves (Figure 1e,f), the cathode polarization potential and the potential difference between 100 and 1000 rpm for the sequential addition of PQ and SPS to volatile metal salt solution (VMS) are shown. In Figure 1e, oscillating polarization occurs when SPS is added. The interaction between PQ and SPS was observed by adding PQ to the solution. The electrode potential decreased rapidly after the addition of PQ, and depolarization occurred over time, likely due to changes in the interaction strength between SPS and PQ. This observation suggests that SPS (sodium polystyrene sulfonate) is linked to the accelerated reduction of Cu by chloride, highlighting a correlation between the electrochemical properties of PQ and its concentration. It is noteworthy that the Δ*η* value is unstable upon the addition of 10 mL/L SPS and stabilizes upon the addition of 20 mL/L PQ, but gradually decreases to 0 over time during subsequent additions of PQ (50–200 mL/L). The difference in PQ adsorption across flow rates was corrected, confirming changes in the convection-dependent adsorption (CDA) of PQ at different concentrations. At low concentrations, the quaternary ammonium cation likely plays a dominant role in adsorption, with SPS interactions leading to an anti-CDA effect. As the concentration increases, the ether group may become the primary driver of adsorption, and the CDA effect diminishes. Thus, the adsorption mode of PQ varies between low and high concentrations.

To investigate the interaction between PQ and SPS, a series of experiments were conducted with varying concentrations of PQ and SPS, as shown in Figure 1f. When 200 mL/L PQ was added, the curve exhibited strong polarization, with a potential difference observed between 100 and 1000 rpm. In the absence of SPS, PQ showed significant polarization and a clear CDA phenomenon, with greater polarization under strong forced convection. At 100 rpm, the potential was −0.65 V compared to −0.68 V at 1000 rpm, suggesting a connection to the adsorption of quaternary ammonium cations in the solution.

Adding 5 mL/L SPS reversed the CDA phenomenon, with a decrease in potential difference under varying speed conditions. With 10 mL/L SPS, the CDA effect disappeared, and the potential difference (Δ*η*) between speeds reduced to zero. This indicates that PQ may change its adsorption mode from cationic adsorption to ether adsorption after adding SPS, which makes the adsorption of PQ on the electrode surface more uniform. The anti-CDA phenomenon reappeared with the sequential addition of 20–40 mL/L SPS, with a more pronounced depolarization effect observed at 1000 rpm compared to 100 rpm. This indicates that PQ exhibits higher inhibition ability under strong forced convection environments without SPS. This phenomenon is related to the transport of Cl^−^ in various convective environments and the adsorption dynamics of both SPS and PQ. The ether group in PQ has minimal influence on the liquid flow rate during adsorption and does not exhibit convection-dependent adsorption (CDA). Therefore, with the addition of SPS, this component may contribute significantly to the adsorption process, stabilizing PQ and maintaining its inhibitory effect. Additionally, quaternary ammonium cations may interact with SPS, leading to anti-CDA phenomena at different rotational speeds. This phenomenon was also confirmed by Li et al. [22], who found that the type of steric hindrance and nitrogen-containing groups (such as tertiary amine or quaternary ammonium with counteranion) in the leveler not only influenced its adsorption ratio and strength on the Cu interface but also affected its antagonistic ability against the accelerator SPS. This suggests that PQ’s suppression ability is better under weak forced convection conditions at the top of the Cu column, compared to strong forced convection conditions.

### 3.2. Quantum Chemical Computations

According to fragment molecular orbital (FMO) theory [23], a higher/highest occupied molecular orbital (HOMO) energy value indicates that the additive is more inclined to donate electrons to the acceptor, while a lower/lowest unoccupied molecular orbital (LUMO) energy value indicates that the additive can accept electrons. As shown in Figure 2, the HOMO is mainly distributed in the ethylene glycol unit, and the LUMO is mainly concentrated in the benzene ring of PQ, indicating that the PQ structure relies on the benzene ring for adsorption. The PQ molecule has an energy gap of 5.2 eV. Notably, according to the hard and soft acid-base (HSAB) principle, a lower ∆E value of the molecule indicates a stronger interaction with Cu (the softest acid). Compared with PEG (∆E~8.28 eV) [24] and Pyridinium (∆E~11.11 eV) [25], PQ has a stronger adsorption capacity [26], all the values are shown in Table 1. Electron spin density (ESP) mapping helps describe the distribution of electron density, whereby high ESP values correspond to low electron densities and are easily adsorbed onto cathode regions with high electron densities. The benzene ring in Figure 2c is in a region with high ESP values, whereas the low ESP distribution with the ether portion suggests that the benzene ring portion of PQ is more susceptible to adsorption on metal surfaces [27]. In Figure 2b, the ESP area distribution diagram shows that the ESP of PQ is mostly distributed around 90 ± 10 kcal/mol, and the region with low electron density is small, indicating that PQ easily adsorbs on the cathode surface.

As shown in Figure 3, the dynamic process of PQ deposition on Cu was simulated to clarify the adsorption mechanism of PQ. The PQ molecules demonstrated stable adsorption capacity during Cu electrodeposition and adhered effectively to the Cu surface. During the adsorption process, the benzene ring of the PQ molecule is partially arranged parallel to the Cu surface, promoting strong surface bonding. According to the simulation results, the benzene ring of PQ has a stronger adsorption effect than the ethylene glycol unit. This explains the CDA behavior of PQ. Essentially, the high rotational speed of the working electrode enhances ion mass transfer to the electrode surface [28]. 

### 3.3. Microstructure and Morphology

Cu pillars electroplated with different SPS and PQ concentrations are proceeded to SEM for observation of the middle of the Cu pillar surface and cross-sectional morphologies (Figure 4 and Figure 5). Smooth and fine-grained plating morphology were achieved under different magnifications for all samples examined, and Cu nodules or other surface defects were eliminated [29], exhibiting good surface coverage and a wide operation window of PQ in the absence of any PEG suppressors. Figure 4(b,b’) show that flatter surfaces were achieved with the presence of 10 mL/L SPS and 200 mL/L PQ in the solution. This supports the observations in Figure 1e,f, indicating that additive adsorption in solutions with 10 mL/L SPS and 200 mL/L PQ is no longer convection-dependent, demonstrating effective leveling of the Cu pillar. In contrast, Figure 4(a,a’,c,c’) reveal that the leveling effect of PQ diminishes when the additive ratio is altered. Combined with Figure 1e,f, it can be inferred that the weakening of the leveling effect is due to the emergence of the anti-CDA phenomenon.

Moreover, cross-sectional profiles of electroplated Cu pillars display varying degrees of doming, i.e., where the central area of a pillar is flat, but the corner part is sharp (Figure 5a,c) or blunt (Figure 5b). It is worth noting that the rotational speeds vary between the edge and the center of the Cu column, with the center experiencing strong forced convection and the edge exhibiting weak forced convection [30]. As observed in Figure 5(a,a’,c,c’), at 10 mL/L SPS with 20 mL/L PQ or 20 mL/L SPS with 200 mL/L PQ, weak forced convection results in stronger inhibition, leading to the formation of a raised shape in the middle of the Cu column. This is supported by the electrochemical data in Figure 1e,f, which shows that anti-CDA effects are more pronounced in the edge region under strong forced convection (1000 rpm), and less effective in the middle region under weak forced convection (100 rpm). The amount of PQ adsorbed in the edge region may be higher due to anti-CDA behavior, resulting in more pronounced anti-CDA effects there compared to the middle region. In solutions with 10 mL/L SPS and 200 mL/L PQ, the anti-CDA phenomenon disappears, and passivation is significantly improved, leading to a better-shaped Cu column with less pronounced anomalies.

The combination of electrochemical, quantum chemistry (QC), and actual electroplating deposition infers that the adsorption strengths of the two PQ structures are different, with the quaternary ammonium salt structure having a stronger adsorption capacity than the ether group. These structures exhibit a synergistic effect under various conditions. At low concentrations, the quaternary ammonium cation structure quickly adsorbs onto the metal surface. When convective adsorption is insufficient, the ether group maintains stable adsorption of PQ on the metal surface, providing inhibition. Consequently, PQ exhibits a more effective inhibition effect under conditions of strong forced convection. When SPS is transferred to an area with strong forced convection, the sulfonic acid group in SPS, carrying a negative charge, interacts with quaternary ammonium cation additives, thereby affecting its inhibition ability. 

### 3.4. Thermal Aging 

Cu pillars electroplated with 10 mL/L SPS and 200 mL/L PQ were selected to prepared SAC305/Cu solder bumps, which were then thermally aged for 96 and 168 h at temperatures of 150 and 200 °C, respectively. The jointing interface on a Cu pillar was examined by SEM, and Kirkendall voids were measured by ImageJ software 1.53t (Figure 6 and Figure 7). Interestingly, as indicated by porosity measurements at randomly selected areas, Kirkendall voids by aging at 150 °C seem to have an obvious, nonuniform distribution: voiding is more serious (>5%) at two of the corners than at the center (1~2%). Generally, it is believed that formation of Kirkendall voids is related to Cu microstructural defects and impurities. A high impurity incorporation results in a more voided jointing interface and the poor reliability of Cu pillar bumps. Since one main impurity source in an electroplating bath is additives, it can be inferred that additive deposition at central and corner areas on the same Cu pillar are different, ascribed to different local additives concentration. It is known that the flow rate at the center of the Cu column in solution is faster than at the edge. According to the electrochemical test in Figure 1, the PQ of the edge part is less affected by SPS. The difference in predominant PQ inhibition ability generates a domed corner morphology. As a result, nonuniform impurity incorporation leads to a nonuniform Kirkendall void distribution. 

Similar nonuniform distributions of Kirkendall voids are also found for aging at 200 °C. Again, voiding is more serious (>9%) at two of the corners than at the center (6~7%), despite smaller numerical differences. Porosity values become higher due to enhanced Cu recrystallization and SAC305/Cu interfacial reaction at an elevated temperature approaching the melting point of SAC305 (217 °C). However, it is hard to isolate the leveler from other effects such as additive interactions, electric field distribution, and flow filed distribution; more experimental and simulation analyses should be carried out for more direct evidence of this mechanism.

### 3.5. Material Characterization

Figure 8 presents and analyzes the XRD patterns and texture coefficients of electroplated Cu pillars with varying concentrations of SPS and PQ. According to PDF card No. 04-0836, diffraction peaks observed at 2theta of 43.297°, 50.4333°, and 74.130° correspond to Cu (111), (200), and (220) crystal planes, respectively. XRD patterns illustrate a higher diffraction intensity and thereby preferred orientation of Cu (111) plane, compared to the randomly oriented Cu powder. This is commonly observed for electroplated Cu with the addition of strong inhibitive additives [31,32,33], which get adsorbed to block high surface-energy Cu (200) and (220) planes, and allow exposure of low surface energy Cu (111) for growth. Cu crystal growth is shown independent of additive concentrations. The texture coeffect ratios for varied SPS and PQ contents exhibit little difference, except for an extreme case with 10 mL/L SPS and 200 mL/L PQ, where the content of suppressive PQ severely exceeds that of accelerating SPS to bring about weakened Cu (111) [34]. This is due to the selective adsorption of the additive in a specific crystal direction. It is noteworthy that these two-additive formulae are generally less preferred than three-additive ones reported elsewhere, which add typical suppressors like PEG. This can be conducive to avoid building strong tensile stress and for improving the ductility of electroplated Cu.

Figure 9 shows the EBSD maps of the cross-section of the central and fringe zones of the Cu column, with the scanning positions illustrated in Figure 9c. Area (a) represents the strong forced convection zone in the central part of the Cu column, while area (b) represents the fringe zone with weak forced convection. Table 2 lists the percentage and length of grain boundaries for both large and small grains at these locations. In Figure 9a,b, a uniform distribution of large and small grain boundaries is observed in the strong forced convection region, whereas the weak convection region shows a clustering of small-angle grain boundaries. This phenomenon suggests an anomalous distribution of grain boundary sizes due to inhomogeneous adsorption of additives in the different convection regions. Figure 9(a’,b’) display corresponding antipodal images, in which the (111) facet appears selectively oriented in both positions. As previously mentioned, the (111) facet has lower surface energy, whereas (101) and (100) have higher surface energies but do not show preferential orientation. This is likely due to selective inhibition by PQ, preventing growth on high-energy facets and promoting low-energy facet orientation. In terms of grain size and orientation difference angle distribution, the central region of the Cu column exhibits small-angle grain boundaries (<15°), large-angle grain boundaries (>15°), and twin grain boundaries with a major orientation difference angle of 60°. In contrast, the fringe region shows an increased frequency of small-angle grain boundaries and a reduced percentage of twin grain boundaries, as shown in Table 2. Specifically, the edge region has 29.1% small-angle grain boundaries and 70.9% large-angle grain boundaries. Although the grain size distribution is not significantly different between the central and edge regions, there are more fine grains in the edge region. This result aligns with the electrochemical curves, indicating that the slower convection rate in the edge region results in the stronger inhibition of PQ formation compared to the central region. Consequently, anomalous additive adsorption leads to refined grain growth in the Cu pillars.

## 4. Conclusion

In summary, a self-synthesized polyquaterntum (PQ) has been studied and formulated with SPS for Cu pillar electroplating on packaging substrates, exhibiting advantages such as a simplified two-additive formulation, high current electroplating, and acceptable domed to flat morphology. In electrochemical tests, PQ exhibited cathodic polarization and convection dependence. The addition of SPS and PQ resulted in an antagonistic reaction. In environments with strong convection, the reaction between SPS and PQ occurred more rapidly than in those with weak convection, leading to a reduction in PQ’s inhibition effect. Conversely, in a convective environment where the reaction was slower, PQ retained its inhibition effect. This suggests that the additive tends to adsorb more at the edge region. Conjugation and QC calculations revealed the synergistic effect between the ether structure and the quaternary ammonium cation structure of PQ during adsorption. The quaternary ammonium cation structure in PQ exhibits convective adsorption dependence during electrodeposition and has a stronger adsorption force than the ether structure, causing it to be adsorbed first on the Cu column surface. With the addition of SPS, the ether structure of PQ, which lacks convective dependence, plays a stronger adsorption role, achieving total absence of convective dependence with a solution ratio of 10 mL/L SPS and 200 mL/L PQ. This was confirmed in subsequent electrodeposition of Cu pillars, where the resulting Cu pillars displayed better flatness at this ratio. The change in adsorption tendency under different convective environments was verified by analyzing the different inhibition states of PQ and SPS under varying convective currents. Based on precise analyses of microstructure and Kirkendall voiding in electroplated Cu pillars and thermally aged SAC305/Cu joint interfaces, it was found that more Kirkendall voids formed at the corners than at the center along the Cu pillar surface. This is closely associated with the relatively higher PQ adsorption compared to SPS, leading to increased SPS breakdown and impurity incorporation at the corners. Meanwhile, EBSD scans demonstrated that finer Cu grains formed at the edge zone, further confirming these results.

## Figures and Tables

**Figure 1 materials-17-05405-f001:**
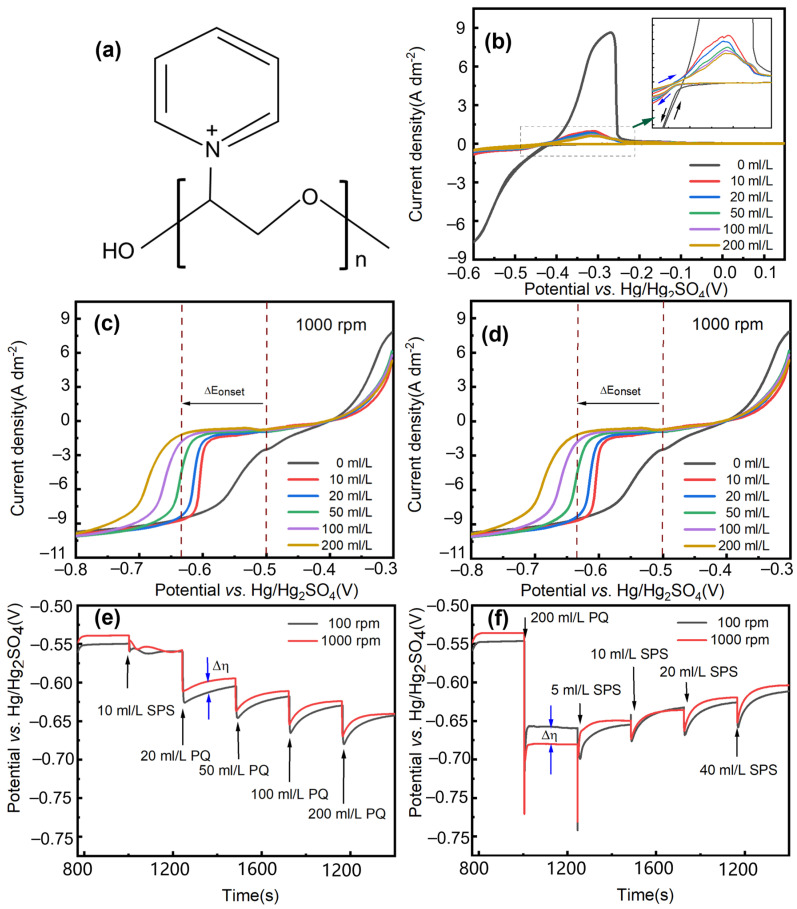
(**a**) Structural formula of PQ. Electrochemical measurements: (**b**) CV curves in presence of 0~200 mL/L PQ; (**c**) LSV curves in presence of 0~200 mL/L PQ, WE rotating speed: 1000 rpm; (**d**) LSV curves in presence of 0~200 mL/L PQ, WE rotating speed: 100 rpm; (**e**) E-t curves at a current density of 5 ASD with sequential addition of 10 mL/L SPS and 10~200 mL/L PQ; (**f**) E-t curves at a current density of 5 ASD with sequential addition of 200 mL/L SPS and 5~40 mL/L SPS.

**Figure 2 materials-17-05405-f002:**
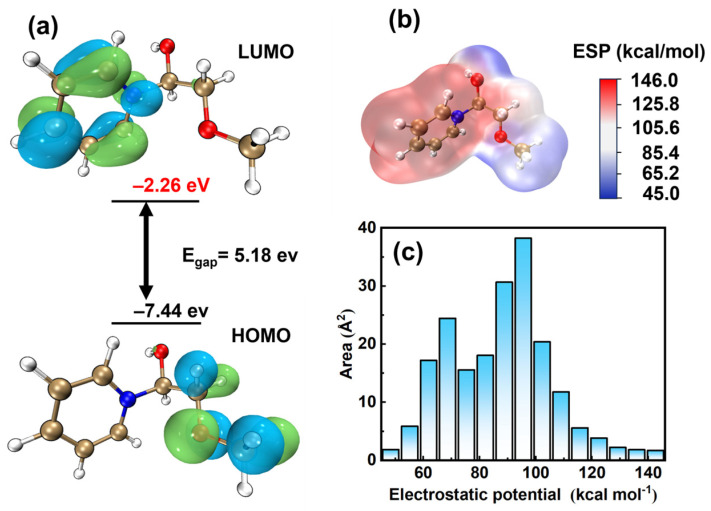
Schematic illustrations of PQ: (**a**) HOMO and LUMO; (**b**) ESP mapping and (**c**) distribution. (N blue, C yellow, O red, H white).

**Figure 3 materials-17-05405-f003:**
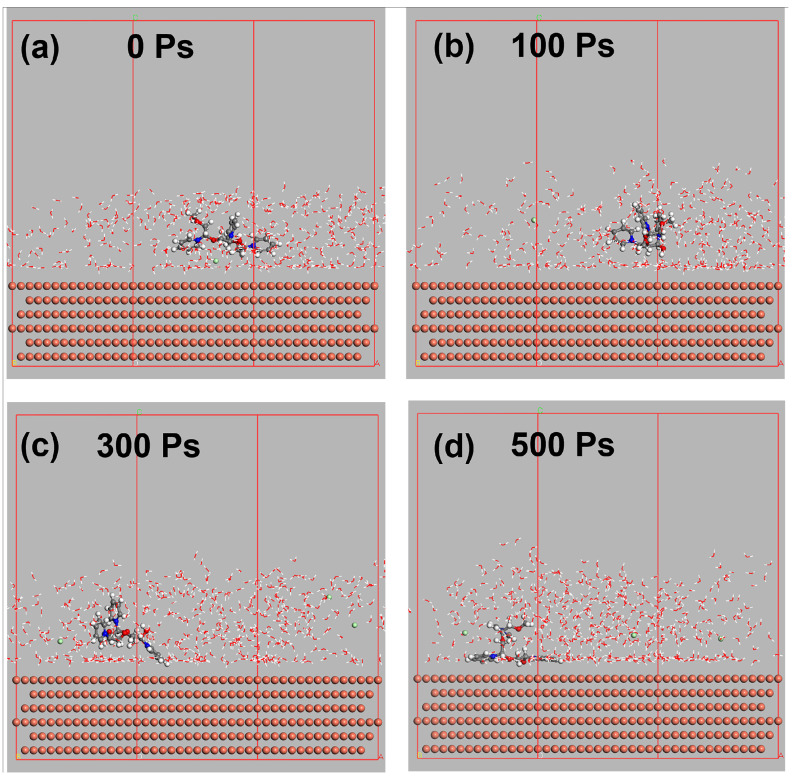
Molecular dynamic simulations of PQ. (**a**) 0 Ps; (**b**) 100 Ps; (**c**) 300 Ps; (**d**) 500 Ps.

**Figure 4 materials-17-05405-f004:**
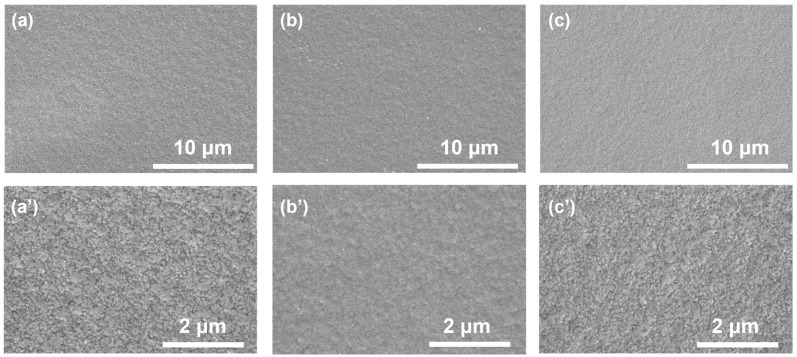
SEM surface morphologies of Cu pillars electroplated at 5 ASD with (**a**,**a’**) 10 mL/L SPS and 20 mL/L PQ; (**b**,**b’**) 10 mL/L SPS and 200 mL/L PQ; (**c**,**c’**) 20 mL/L SPS and 200 mL/L PQ. (**a**–**c**) magnification ×5000; (**a’**–**c’**) magnification ×20,000.

**Figure 5 materials-17-05405-f005:**
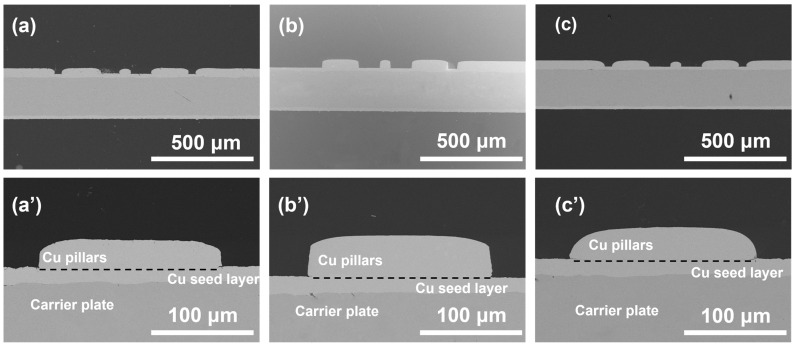
SEM cross-sectional morphologies of Cu pillars electroplated at 5 ASD with (**a**,**a’**) 10 mL/L SPS and 20 mL/L PQ; (**b**,**b’**) 10 mL/L SPS and 200 mL/L PQ; (**c**,**c’**) 20 mL/L SPS and 200 mL/L PQ. (**a**–**c**) magnification ×100; (**a’**–**c’**) magnification ×500.

**Figure 6 materials-17-05405-f006:**
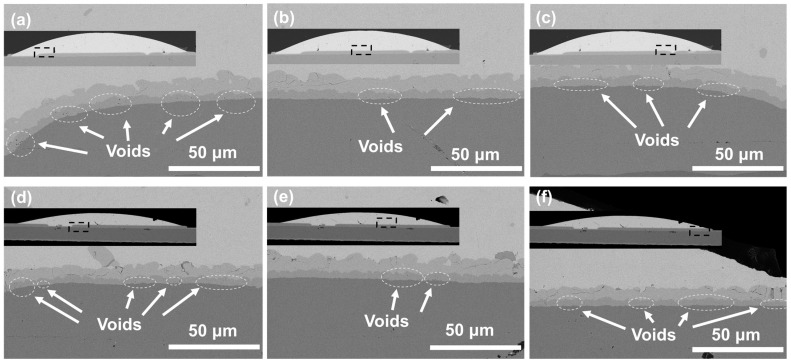
SEM cross-sectional morphologies of SAC305/Cu jointing interfaces on a Cu pillar bump after thermal aging at 150 °C for (**a**–**c**) 96 h and (**d**–**f**) 168 h.

**Figure 7 materials-17-05405-f007:**
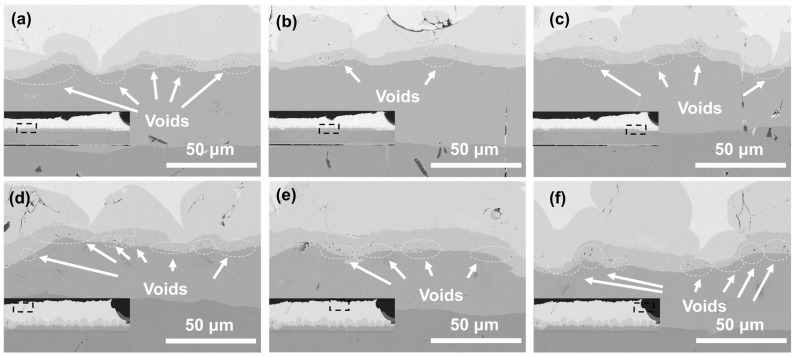
SEM cross-sectional morphologies of SAC305/Cu jointing interfaces on a Cu pillar bump after thermal aging at 200 °C for (**a**–**c**) 96 h and (**d**–**f**) 168 h.

**Figure 8 materials-17-05405-f008:**
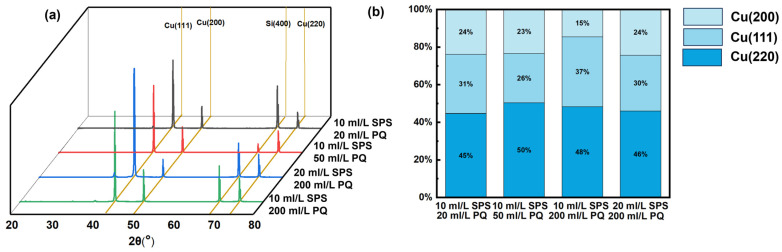
(**a**) XRD patterns of Cu films electroplated with different electroplating baths; (**b**) texture coefficients plots.

**Figure 9 materials-17-05405-f009:**
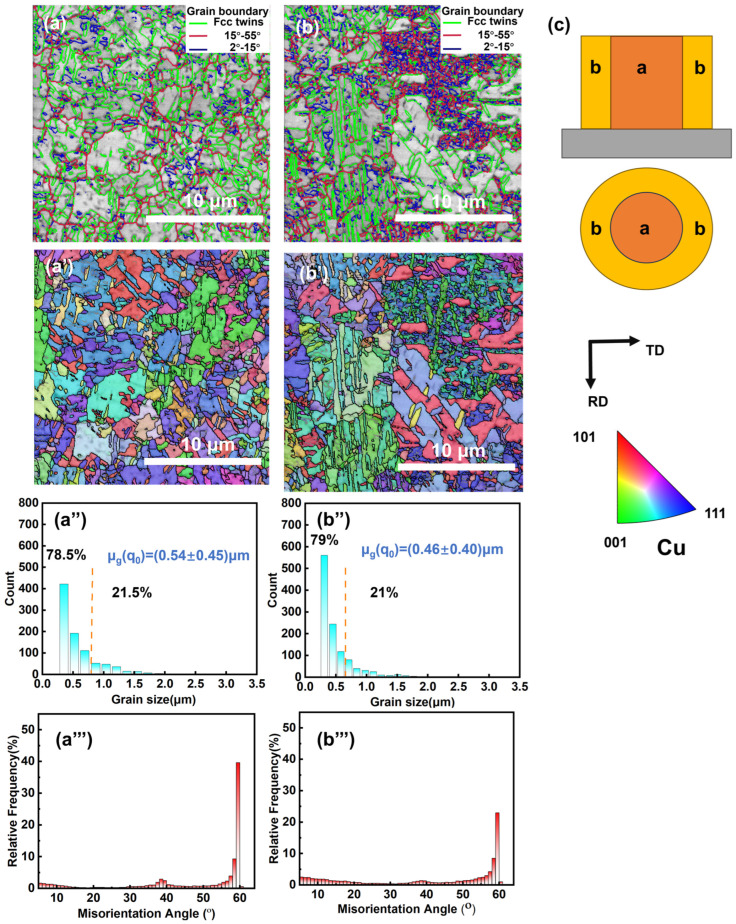
EBSD test diagram: (**a**,**b**) BC diagrams at different locations; (**a’**,**b’**) corresponding grain orientation diagram; (**a’’**,**b’’**) corresponding grain size distribution map; (**a’’’**,**b’’’**) grain boundary angle distribution; (**c**) shooting position diagram. EBSD results were obtained at the normal direction (ND) of the samples.

**Table 1 materials-17-05405-t001:** Statistical data of grain boundaries of the Cu pillar at different positions.

Additive	LUMO (eV)	HOMO (eV)	Gap (eV)
PQ	−2.26	−7.44	5.18
PEG	1.07	−7.21	8.28
Pyridinium	−1.01	−12.12	11.11

**Table 2 materials-17-05405-t002:** Statistical data of grain boundaries of Cu pillar at different positions.

Position	Area Fraction of LAGB (%)	Area Fraction of HAGB (%)	Area Fraction of TA (%)	Grain Boundary Length (mm)
Central region	17.5	23.1	59.4	1496.6
Outer ring region	29.1	24.5	46.3	1883.9

Note: LAGB—low-angle grain boundary; HAGB—high-angle grain boundary; TA—Twin Angle.

## Data Availability

The original contributions presented in the study are included in this article; further inquiries can be directed to the corresponding author.

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
