# Peer review of "Cu Pillar Electroplating Using a Synthetic Polyquaterntum Leveler and Its Coupling Effect on SAC305/Cu Solder Joint Voiding"

_materials, 2024, doi:10.3390/ma17225405_

Round 1
Reviewer 1 Report
Comments and Suggestions for Authors
The title of the work is not clear, it is advisable to define according to the relevance of the proposed study. This is considered security.
The relevance of the study is identified and supported by an adequate analysis of the state of the art. The novelty is not clear, it is understood that it is oriented to a self-synthesized process for electroplating. The effect or relevance of additives is not clear. Clarifying the novelty in detail is recommended.
For the electrochemical study, the study of variables of relevance with respect to cathodic polarization and convection, it is advisable to go deeper. As well as the rationale for the change in the PQ adsorption mechanism, and why it is concluded that the addition of SPS led to stronger inhibition. It seems that additives play a very relevant role and the scientific support in this regard is advisable to be deeper and more robust.
3 Figure quality, can be improved.
The discussion and analysis of results is adequate and supported by scientific literature.
It is advisable to show SEM (Figure 4) at higher magnification. In order to see in detail the morphology of the material. Since it seems to play a relevant role. Perhaps a particle size histogram or roughness data might be suitable.
Figure 5 scale is correct, comparing with Figure 5, it considers an order of magnitude difference. Clarification in this regard is recommended.
Author Response
Manuscript ID: materials-3246829
Title: Cu pillar electroplating using a synthetic polyquaterntum leveler and effect of it and SPS on the porosity of SAC305/Cu solder joint at different flow rates
Dear Editor,
Thanks for the decision letter. We appreciate very much the time you and the reviewers have spent in carefully examining our manuscript and the helpful comments. We have revised the manuscript to fully address the comments from you and the reviewers. The revised manuscript has been uploaded to the journal website with all changes made highlighted in yellow. Please see below for our detailed responses to comments.
- The title of the work is not clear, it is advisable to define according to the relevance of the proposed study. This is considered security.
We thank the editor for pointing out the problem about conduction. In response, we have added a section on electroplating solder joints in the introduction.
” Micro-bumps in three-dimensional integrated circuits (3D ICs) have emerged as a crucial technology. As the miniaturization of power devices approaches its limits, 3D packaging technology offers a viable solution for achieving higher density. However, this advancement has also led to significantly heightened requirements for micro-bumps[12]. The electrodeposition process frequently yields a highly reliable solder layer. Szostak et al[13] reported that by the electrodeposition of Au-Sn eutectic, a distinct pattern is formed, resulting in a bonding strength exceeding 70%. However, usually, improving the solder joint reliability of electroplated copper materials substantially depend on additive formulae and electroplating parameters, which determine the growth morphology, crystal structure and impurity incorporation in the process of electrodeposition and electroplating parameters [14,15]. Chiang et al [16] reported Cu electroplated films be impurities affected. That after thermal 200℃ Ageing many voids appear in the interface, due to huge sum of impurities incorporated in the Cu electroplated films. The control of additive and the quantity of impurities in Cu electroplated films significantly affect void formation at the joint interface. However, relative investigating about these factors is rare. ”
- The relevance of the study is identified and supported by an adequate analysis of the state of the art. The novelty is not clear, it is understood that it is oriented to a self-synthesized process for electroplating. The effect or relevance of additives is not clear. Clarifying the novelty in detail is recommended.
We thank the editor suggestion to highlight the novelty and discuss more about the role of additives. In response, we have detailed the action and emphasized that the self-synthesizing additive is synthesized from two different acting additives and has the action of both additives.
“We developed a quaternary ammonium salt additive. It has new additives structure with those of leveler and suppressor. As depicted in Fig. 1(a), the PQ structure of the additive comprises two parts: the benzene ring structure of the positively charged quaternary ammonium cation and the ether. PEG is a very typical class of inhibitors whose main structure is an ether [18], also quaternary ammonium salt cationic additives have excellent inhibition and leveling effects [19]. Jo et al [20] again suggested that the hydrophobic tail of the quaternary ammonium salt inhibits Cu deposition, while the cationic portion adsorbs the anions on the surface of the Cu and thus reduces the roughness of the Cu electrodeposition. To achieve an additive with both corrosion inhibition and leveling effect. Therefore, we developed a quaternary ammonium salt additive. This additive is a combination of the ether of PEG and the benzene ring and cation in the quaternary ammonium salt.”
- For the electrochemical study, the study of variables of relevance with respect to cathodic polarization and convection, it is advisable to go deeper. As well as the rationale for the change in the PQ adsorption mechanism, and why it is concluded that the addition of SPS led to stronger inhibition. It seems that additives play a very relevant role and the scientific support in this regard is advisable to be deeper and more robust.
We thank the editor for suggesting that we deepen our research in electrochemistry and explain the effects of SPS and PQ in the same environment. In response, we show that the antagonism between SPS and PQ occurs, and the reaction between SPS and PQ is fast and the inhibition and leveling effects are weakened at the fast flow rate. Where the flow rate is slow, the reaction is slower, and the PQ throw maintains the inhibiting and leveling effect.
“The difference in polarization resulting from changes in flow rate can be attributed to the rapid reaction of the quaternary ammonium cation with Cu, which produces an inhibition effect in high-flow conditions. Conversely, at a slow flow rate, such as 100 rpm, the diffusion of additives is sluggish, limiting their effectiveness. As a result, the polarization interval at 100 rpm is narrower than that observed at 1000 rpm.”
“This indicates that PQ may change its adsorption mode from cationic adsorption to ether adsorption after adding SPS, which makes the adsorption of PQ on the electrode surface more uniform.”
“In environments with strong convection, the reaction between SPS and PQ occurred more rapidly than in those with weak convection, leading to a reduction in PQ's inhibition effect. Conversely, in a convective environment where the reaction was slower, PQ retained its inhibition effect. This suggests that the additive tends to adsorb more at the edge region. Conjugation and QC calculations revealed the synergistic effect between the ether structure and the quaternary ammonium cation structure of PQ during adsorption. The quaternary ammonium cation structure in PQ exhibits convective adsorption dependence during electrodeposition and has a stronger adsorption force than the ether structure, causing it to be adsorbed first on the Cu column surface.”
- 3 Figure quality, can be improved.
We thank the editor for pointing out the problem about image quality. In response, we have improved the quality of Figure 3.
- The discussion and analysis of results is adequate and supported by scientific literature.
We are pleased with the editor's comments.
- It is advisable to show SEM (Figure 4) at higher magnification. In order to see in detail the morphology of the material. Since it seems to play a relevant role. Perhaps a particle size histogram or roughness data might be suitable.
We appreciate the editor's suggestion to magnify the surface of Cu with a higher magnification, as well as the use of other roughness experiments for evaluation. But we have increased the scan magnification to 20,000 times, which is close to the limit of magnification. The rough state of the material is obvious. As others have reported ”Effect of Cu(I) on electrochemical behavior and surface morphology of electrodeposited copper for different accelerators”
- Figure 5 scale is correct, comparing with Figure 5, it considers an order of magnitude difference. Clarification in this regard is recommended.
Thanks to the editor's suggestion, we have written the magnification in the article.
Figure 5. SEM cross-sectional morphologies of Cu pillars electroplated at 5 ASD with (a and a') 10 ml/L SPS and 20 ml/L PQ; (b and b') 10 ml/L SPS and 200 ml/L PQ; (c and c') 20 ml/L SPS and 200 ml/L PQ. (a-c) magnification x 100; (a’- c’) magnification x500.
Thanks for your kind editorial help.
Sincerely,

Reviewer 2 Report
Comments and Suggestions for Authors
The authors have suggested that an alternative soldering and packaging alternative, the methodology and experimental sections are well organized. However, there are a few issues that need to be addressed before suggesting for publication.
The introduction needs to be further strengthened by providing more examples on the soldering aspect of this project e.g., 10.1088/1361-6439/ac12a1
Many abbreviations have not been properly explained, please correct.
Could the authors provide elemental characterization of the electroplated surface using EDX elemental mapping?
Could the authors comment on the adhesion of the electroplated Cu on the sublayer?
Author Response
Manuscript ID: materials-3246829
Title: Cu pillar electroplating using a synthetic polyquaterntum leveler and effect of it and SPS on the porosity of SAC305/Cu solder joint at different flow rates
Dear Editor,
Thanks for the decision letter. We appreciate very much the time you and the reviewers have spent in carefully examining our manuscript and the helpful comments. We have revised the manuscript to fully address the comments from you and the reviewers. The revised manuscript has been uploaded to the journal website with all changes made highlighted in yellow. Please see below for our detailed responses to comments.
1.The authors have suggested that an alternative soldering and packaging alternative, the methodology and experimental sections are well organized. However, there are a few issues that need to be addressed before suggesting for publication.
We thank the editor for pointing out the problem about conduction. In response, we have added a section on electroplating solder joints in the introduction.
“Micro-bumps in three-dimensional integrated circuits (3D ICs) have emerged as a crucial technology. As the miniaturization of power devices approaches its limits, 3D packaging technology offers a viable solution for achieving higher density. However, this advancement has also led to significantly heightened requirements for micro-bumps[12]. The electrodeposition process frequently yields a highly reliable solder layer. Szostak et al.[13] reported that by the electrodeposition of Au-Sn eutectic, a distinct pattern is formed, resulting in a bonding strength exceeding 70%. However, usually, improving the solder joint reliability of electroplated Cu materials substantially depend on additive formulae and electroplating parameters, which determine the growth morphology, crystal structure and impurity incorporation in the process of electrodeposition and electroplating parameters [14,15]. Chiang et al. [16] reported Cu electroplated films be impurities affected. That after thermal 200℃ Ageing many voids appear in the interface, due to huge sum of impurities incorporated in the Cu electroplated films. The control of additive and the quantity of impurities in Cu electroplated films significantly affect void formation at the joint interface. However, relative
2.The introduction needs to be further strengthened by providing more examples on the soldering aspect of this project e.g., 10.1088/1361-6439/ac12a1
We thank the editor’s suggestion that we cite more relevant cases. In response,we added relevant references, which are helpful to understand the topic of this work:
- Y, W, Chang.; C, C, Hu.; H, Y, Peng.; Y, C, Liang.; C, Chen.; T, C, Chang.; J, Y, Juang. A new failure mechanism of electromigration by surface diffusion of Sn on Ni and Cu metallization in microbumps. Scientific reports, 2018, 8(1), 5935.
- K, M, Szostak.; M, Keshavarz.; T, G, Constandinou. Hermetic chip-scale packaging using Au: Sn eutectic bonding for implantable devices. Journal of Micromechanics and Microengineering, 2021, 31(9), 095003.
- H, Lee.; S, T, Tsai.; P, H, Wu. Influence of additives on electroplated copper films and their solder joints. Materials Characterization, 2019, 147, 57-63.
- T, Y, Kim.; S, Choe.; J, J, Kim. Decomposition of polyethylene glycol (PEG) at Cu cathode and insoluble anode during Cu electrodeposition. Electrochimica Acta, 2020, 357, 136803.
- P, C, Chiang.; Y, A, Shen.; C, M, Chen. Effects of impurities on void formation at the interface between Sn-3.0 Ag-0.5 Cu and Cu electroplated films. Journal of Materials Science: Materials in Electronics, 2021,32, 11944-11951.
3.Many abbreviations have not been properly explained, please correct.
Thanks to the editor for the reminder. In response, we added a lot of shorthand explanations to the article.
” self-synthesized polyquaterntum (PQ)”
“bis- (sodium sulfopropyl)-disulfide (SPS)”
“polyethylene glycol (PEG)”
“Janus green B (JBG)”
” cyclic voltammetry (CV)”
” counter electrode (CE)“
”reference electrode (RE)”
” Texture coefficients (TC)”
“convection-dependent adsorption (CDA)”
” hard and soft acid-base (HSAB)”
” fragment molecular orbital (FMO)”
”higher highest occupied molecular orbital (HOMO)”
“energy lower lowest unoccupied molecular orbital (LUMO) energy”
4.Could the authors provide elemental characterization of the electroplated surface using EDX elemental mapping?
Thanks to the editor for suggestion. In fact, EDS detection of Cu coating cannot fully prove the state of impurities in Cu layer, since they are too less to be detected accurately according to EDS. In our group’s previous work, the impurities in Cu layer has been investigated by SIMS, as shown in reference 11” Effects of suppressors on the incorporation of impurities and microstructural evolution of electrodeposited Cu solder joints” of the revised manuscript. We cite this article to support the conclusion in this manuscript.
6.Could the authors comment on the adhesion of the electroplated Cu on the sublayer?
This work mainly focuses on the electrodeposited shape of the copper pillar itself and the formation of aging Kirkendall holes at the interface. We think the adhesion of Cu pillar on the substrate should be no different from other works.
Thanks for your kind editorial help.
Sincerely,
